# Identification of GM1-Ganglioside Secondary Accumulation in Fibroblasts from Neuropathic Gaucher Patients and Effect of a Trivalent Trihydroxypiperidine Iminosugar Compound on Its Storage Reduction

**DOI:** 10.3390/molecules29020453

**Published:** 2024-01-17

**Authors:** Costanza Ceni, Francesca Clemente, Francesca Mangiavacchi, Camilla Matassini, Rodolfo Tonin, Anna Caciotti, Federica Feo, Domenico Coviello, Amelia Morrone, Francesca Cardona, Martino Calamai

**Affiliations:** 1Department of Chemistry “U. Schiff” (DICUS), University of Florence, Via della Lastruccia 3-13, 50019 Sesto Fiorentino, Italy; ceni@lens.unifi.it (C.C.); francesca.mangiavacchi@unifi.it (F.M.); camilla.matassini@unifi.it (C.M.); francesca.cardona@unifi.it (F.C.); 2European Laboratory for Non-Linear Spectroscopy (LENS), University of Florence, 50019 Sesto Fiorentino, Italy; 3Laboratory of Molecular Biology of Neurometabolic Diseases, Neuroscience Department, Meyer Children’s Hospital IRCCS, 50139 Florence, Italy; rodolfo.tonin@meyer.it (R.T.); anna.caciotti@meyer.it (A.C.); federica.feo@meyer.it (F.F.); amelia.morrone@meyer.it (A.M.); 4Laboratory of Human Genetics, IRCCS Istituto Giannina Gaslini, 16147 Genoa, Italy; domenicocoviello@gaslini.org; 5Department of Neurosciences, Psychology, Drug Research and Child Health (NEUROFARBA), University of Florence, 50121 Florence, Italy; 6National Institute of Optics-National Research Council (CNR-INO), 50019 Sesto Fiorentino, Italy

**Keywords:** metabolic disorders, lysosome, glucocerebrosidase, glucosylceramide, GM1, flow cytometry, pharmacological chaperones

## Abstract

Gaucher disease (GD) is a rare genetic metabolic disorder characterized by a dysfunction of the lysosomal glycoside hydrolase glucocerebrosidase (GCase) due to mutations in the gene GBA1, leading to the cellular accumulation of glucosylceramide (GlcCer). While most of the current research focuses on the primary accumulated material, lesser attention has been paid to secondary storage materials and their reciprocal intertwining. By using a novel approach based on flow cytometry and fluorescent labelling, we monitored changes in storage materials directly in fibroblasts derived from GD patients carrying N370S/RecNcil and homozygous L444P or R131C mutations with respect to wild type. In L444P and R131C fibroblasts, we detected not only the primary accumulation of GlcCer accumulation but also a considerable secondary increase in GM1 storage, comparable with the one observed in infantile patients affected by GM1 gangliosidosis. In addition, the ability of a trivalent trihydroxypiperidine iminosugar compound (CV82), which previously showed good pharmacological chaperone activity on GCase enzyme, to reduce the levels of storage materials in L444P and R131C fibroblasts was tested. Interestingly, treatment with different concentrations of CV82 led to a significant reduction in GM1 accumulation only in L444P fibroblasts, without significantly affecting GlcCer levels. The compound CV82 was selective against the GCase enzyme with respect to the β-Galactosidase enzyme, which was responsible for the catabolism of GM1 ganglioside. The reduction in GM1-ganglioside level cannot be therefore ascribed to a direct action of CV82 on β-Galactosidase enzyme, suggesting that GM1 decrease is rather related to other unknown mechanisms that follow the direct action of CV82 on GCase. In conclusion, this work indicates that the tracking of secondary storages can represent a key step for a better understanding of the pathways involved in the severity of GD, also underlying the importance of developing drugs able to reduce both primary and secondary storage-material accumulations in GD.

## 1. Introduction

Gaucher disease (GD) is the most common lysosomal storage disorder (LSD) of glycosphingolipids. It occurs at a frequency of 1 in 100,000 live births in the general population and of 1 in 450 live births among Ashkenazi Jews [1,2]. GD is caused by mutations in the *GBA* gene (mapped on chromosome 1q21-22), which encodes for the lysosomal enzyme acid β-glucosidase (glucocerebrosidase, also known as GCase and GBA, EC 3.2.1.45, MIM*606463). GCase is an enzyme involved in the lysosomal breakdown of two sphingolipids, glucocerebroside (GlcCer) and glucosylsphingosine (GlcSph). Human GCase comprises three interconnected domains: Domain I (residues 1–27 and 383–414); Domain II (residues 30–75 and 431–497), which resembles an immunoglobulin (Ig) fold; and Domain III (residues 76–381 and 416–430), which contains the catalytic site. Severe mutations in the *GBA* gene are distributed throughout the three domains of the GCase [3]. The severe neuropathic GD phenotypes may arise from a broad range of enzyme defects involving the catalytic site (directly and indirectly), protein folding and/or stability, or regulatory functions, such as the interactions with saposin C or lysosomal-membrane phospholipids [3]. GD is classified into three major clinical types based on age of onset and neurological involvement. GD type 1 is characterized by the presence of clinical or radiographic evidence of bone disease (osteopenia, focal lytic or sclerotic lesions, and osteonecrosis), hepatosplenomegaly, anaemia and thrombocytopenia, lung disease, and the absence of primary central-nervous-system disease. GD types 2 and 3 are characterized by the presence of primary neurologic disease. GD type 2 (acute neuronopathic) is characterized by childhood onset, with aggressive involvement of the central nervous system, often resulting in death within 2 years. GD type 3 (subacute chronic neuronopathic) is characterized by later onset and a slower progressive course [4]. Additional data on the role of each mutation in determining specific GD phenotypes are needed.

Currently available treatments for GD, especially for type 1, include enzyme replacement therapy (ERT) [5] using imiglucerase (Cerezyme^®^, Genzyme, Cambridge, MA, USA), velaglucerase alfa (VPRIV^®^, Shire, Lexington, MA, USA), and taliglucerase alfa (Elelyso^®^ Pfizer, New York, NY, USA) and substrate reduction therapy (SRT) [6] using *N*-butyldeoxynojirimycin (NB-DNJ; Zavesca, Phoenix AZ, USA). The goal of both treatments is to reduce GlcCer storage, thus reducing the detrimental effects caused by its accumulation. ERT achieves this by supplementing defective enzymes with active enzymes, whereas SRT works by lowering the rates of synthesis of all GlcCer, thus reducing glycolipid accumulation. These therapies have different disadvantages, such as the dependence of ERT on intravenous infusions usually every 2 weeks and poor delivery to bones and lungs and its inability to cross the blood–brain barrier [7]. By contrast, Zavesca^®^ is given orally and crosses the blood–brain barrier but causes several side effects [7].

More recently, an alternative therapeutic strategy has emerged, namely the pharmacological chaperone therapy (PCT). Pharmacological chaperones (PCs) have been studied as a potential treatment for GD, particularly in cases where the mutations result in misfolded GCase that retains some residual enzymatic activity. The PCs can interact with the misfolded GCase, stabilizing and helping it fold into its proper conformation. This allows the enzyme to be trafficked from the endoplasmic reticulum to lysosomes and resume its enzymatic activity, thereby reducing the accumulation of GlcCer. PCs behave as reversible inhibitors of the enzyme, but when they are used in sub-inhibitory amounts, they can correct the folding and/or stabilize the enzyme’s catalytic activity with minimal side effects [8,9]. Trihydroxypiperidine iminosugars, carbohydrate analogues in which a nitrogen atom replaces the endocyclic oxygen, have shown interesting properties as anti-viral, anti-bacterial, anti-diabetes, or immunosuppressant agents [10] and, more recently, also as PCs for LSDs [11,12]. In the past twenty years, other piperidine iminosugars have been investigated to this aim, such as 1-deoxygalactonojirimycin (DGJ), which was the first oral PC commercially approved for Fabry disease (another LSD) in Europe (Migalastat, Galafold, Amicus Therapeutics, Philadelphia, PA, USA) [13]. However, no PC for Gaucher disease has yet reached the market. 

The lack of a treatment that targets the primary cause for the neuropathic GD phenotypes has prompted us to dig deeper in the mechanisms involving the first events in the pathogenesis and take into account secondary accumulated materials. Although noted for the first time 40 years ago, few studies have looked at the secondary accumulation in the central nervous system and peripheral organs of GD patients. Higher levels of lactosylceramide (LacCer) were found in the brain of patients affected by different types of GD, while the concentration of gangliosides was found to be normal but with a higher proportion of GM2 and GM3 [14]. An increase in secondary GD3 was found in the brain of a type 2 patient [15]. Ceramide, dihexosylceramide, trihexosylceramide, and phosphatidylglycerol were found to be higher in skin fibroblasts from patients with GD types 1 and 2 [16]. A clear connection between accumulation of GluCer or GluSph and secondary compounds has not yet been clearly established.

In this work, we show that GM1 is also accumulated in skin fibroblasts from GD types 2 and 3 by using a novel approach based on flow cytometry and fluorescent labelling. No known genetic defects were found in beta-Galactosidase (β-Gal), the lysosomal hydrolase responsible for the degradation of GM1, which is located several steps upstream in the catabolic pathway of GlcCer (Table 1). Moreover, the ability of a trivalent trihydroxypiperidine iminosugar compound (CV82), which previously showed good pharmacological chaperone activity on GCase [17] (Table 2), to reduce the levels of both GlcCer and GM1 was evaluated. Our results on the secondary accumulation of GM1 are important for the characterization of the downstream pathways in GD and potentially paving the way for testing β-galactosidase as an alternative therapeutic target for neuronopathic variants of GD with unmet clinical needs.

## 2. Results

### 2.1. Increased GlcCer and GM1 Contents in L444P Fibroblasts Highlighted by Confocal-Microscopy and Flow-Cytometry Analyses

We monitored changes in storage materials in fibroblasts derived from a homozygous GD patient carrying the c.1448T>C (L444P) mutation and from a compound-heterozygous patient carrying the c.1226A>G (N370S)/RecNcil mutations with respect to the wild-type (WT) fibroblasts. While patients who are compound heterozygous for the N370S mutation are linked to type 1 GD, patients homozygous for the L444P mutation are prone to develop type 3 GD [18]. The N370S mutation is located at the interface of Domains II and III, too far from the catalytic site to affect it. The L444P mutation perturbs the stability of the hydrophobic core of Domain II, generating an unstable GCase [3].

In order to label GlcCer, the primary storage material in GD, fibroblasts were first fixed, permeabilized, labelled with a specific primary anti-glucosylceramide antibody coupled to a secondary fluorescent antibody, and imaged with a confocal microscope (Figure 1A). While in the case of the N370S mutant, a slight increase in fluorescence intensity was not found to be statistically significant with respect to the WT, considerable higher levels (~2-fold increase) of labelled GlcCer were detected in L444P fibroblasts.

In addition to GlcCer, the accumulation of GM1 ganglioside as a secondary storage material was examined by using CTXb-FITC, the fluorescent b subunit of the cholera toxin that specifically and strongly binds to GM1 [19] (Figure 1B). Confocal-microscopy analysis showed a ~5-fold increase in the cellular content of GM1 of L444P fibroblasts with respect to WT, which was not observed in N370S/RecNcil fibroblasts. Moreover, the distribution of GM1 in L444P cells was reminiscent of the one observed in GM1 gangliosidosis patients [20]. 

These results were further corroborated by flow-cytometry analysis. In fibroblasts of patients carrying the L444P mutation, a ~50% increase in GlcCer accumulation (Figure 2A) was observed, accompanied by a ~60% increase in GM1 storage (Figure 2B), comparable to the one observed in patients affected by GM1 gangliosidosis [20,21]. On the other hand, the contents of GlcCer and GM1 in fibroblasts of patients carrying the N370S mutation did not vary significantly compared to the control (Figure 2A,B). 

### 2.2. Decreased Accumulation of GM1 in L444P Fibroblasts Treated with CV82

We tested the ability of the CV82 compound (Table 2), a pharmacological chaperone that previously showed an appreciable effect on the GCase enzyme (2-fold activity enhancement at 10 μM) [17], to reduce the levels of storage materials in fibroblasts isolated from patients carrying the L444P mutation. Primary cultures of L444P fibroblasts were incubated for 4 days with three different concentrations of CV82 (10 µM, 1 µM, and 100 nM) and labelled as reported above. Interestingly, treatment with 10 µM and 100 nM of CV82 led to a significant reduction in GM1 accumulation (Figure 3B) without, however, affecting GlcCer levels (Figure 3A). In particular, the maximum decrease in accumulated GM1 was observed at 10 µM and was around 20%.

### 2.3. Unchanged β-Gal Enzyme Activity in WT and Mutated Fibroblasts from Juvenile GM1 Patients Treated with CV82

In order to investigate the mechanisms through which CV82 could decrease the build-up of GM1 without significantly affecting GlcCer levels, we initially hypothesized a direct interaction between the compound CV82 and the β-Gal enzyme responsible for the catabolism of GM1 ganglioside. For this reason, the inhibition and pharmacological chaperone activities of CV82 towards the β-Gal enzyme were evaluated. First of all, we evaluated the inhibition ability of CV82 towards the β-Gal enzyme at 1 mM in human leukocyte homogenates derived from healthy donors. The compound CV82 showed a negligible percentage of inhibition (33% at 1 mM; Figure 4A), demonstrating its complete selectivity against the GCase enzyme with respect to the β-Gal enzyme. It is worth noting that the best inhibitory activity does not always correspond to the best chaperone activity on cell lines, as already observed for a series of compounds [22]. Therefore, we tested the ability of the CV82 compound to increase the β-Gal enzyme activity in the fibroblasts wild type (Figure 4B) and with the p.Arg201His/p.Tyr83LeufsX8 and p.Arg201His/p.Ile51Asn mutations from juvenile GM1 patients (Figure 4C,D). No enzymatic-activity increase was observed in wild-type and mutated fibroblasts after incubation with increasing concentrations of CV82 from 10 nM to 10 μM. The assay could not be performed at a higher concentration (50 μM), since the low cell viability observed hampered the measurement of the enzymatic activity. 

### 2.4. Increased Accumulation of GlcCer and GM1 in R131C Fibroblasts Treated with CV82

Additionally, we evaluated the effect of CV82 on fibroblasts carrying, at the homozygous state, the c.508C>T (R131C) mutation, which has been proven to be an extremely severe mutation that alters the stability and activity of GCase [23,24]. First of all, changes in primary and secondary storage materials in R131C fibroblasts, with respect to the WT control, were monitored by using flow-cytometry analysis (Figure 5), revealing not only an increase (~20%) in GlcCer storage (Figure 5A) but also a substantial (~110%) increase in GM1 levels (Figure 5B). Subsequently, R131C fibroblasts were treated with three different concentrations of CV82 by using the same procedure reported above. Contrary to what had been observed for the L444P mutant, treatment with CV82 led to a significant accumulation of both GlcCer (Figure 5C) and GM1 (Figure 5D).

## 3. Discussion

The disease severity of LSDs is often correlated with the residual activity levels of the mutated enzyme involved. In vitro enzymatic-activity tests use soluble synthetic fluorogenic substrates in cell lysates, which may not take into account potential enzyme activators and may deliver inaccurate results that may not match the clinical course of the disease. We have recently developed an approach based on specific fluorescence labelling coupled to microscopy and flow cytometry capable of detecting higher levels of accumulated metabolites (e.g., sialic acid, GM1 ganglioside, and cholesterol) in fibroblasts from patients affected by several LSDs, such as GM1 gangliosidosis, Niemann–Pick type C, and sialidosis [20,21]. These methods allow one to directly correlate the amount of accumulated material with the severity of the disease. We previously showed that our approach also turns out to be very convenient to rapidly screen the effects of new pharmacological compounds on the accumulation of catabolites directly in treated cells [20]. 

Here, our method was extended to fibroblasts from GD patients, not limiting our experiments to measure the level of GlcCer but also that of GM1 (Figure 1, Figure 2 and Figure 3), and to investigate the effects arising from the treatment of the cells with a previously developed PC directed against GCase [17]. The levels of GlcCer were found to be significantly higher in homozygous-patient fibroblasts affected by the mutations of L444P, which strongly affects the stability of GCase by altering the retention of mutated GCase in the endoplasmic reticulum [25], and R131C, which severely alters the active site structure/function of GCase [23]. It is largely known that the L444P mutation has been directly linked to Gaucher type 2 and type 3 [26,27]. The R131C mutant allele was also identified in type 2 (acute neuronopathic) GD [23,28]. Surprisingly, the levels of GM1 also appeared to be dramatically increased in fibroblasts bearing these mutations, up to the levels observed for infantile patients affected by GM1 gangliosidosis [20,21]. Little or no changes in the GlcCer and GM1 amounts were found for the fibroblasts with the N370S mutation, which is associated with GD type 1. N370S has been proven to be a folding mutant in part retained in the endoplasmic reticulum [29]. Our results show that secondary GM1 accumulation is not negligible, and its importance should not be undervalued in the pathological context. Interestingly, the correlation between the primary GlcCer and secondary GM1 accumulation was found to be not very strong, with the amount of GM1 rather linked to the severity of the disease (Figure 5E).

Interestingly, treatment with different concentrations of the trivalent trihydroxypiperidine iminosugar compound CV82 led to a significant reduction in GM1 accumulation without significantly affecting GlcCer levels in L444P fibroblasts. In order to investigate the mechanisms through which CV82 could decrease the build-up of GM1 without significantly affecting GlcCer levels, a direct interaction between the compound CV82 and the β-Gal enzyme responsible for the catabolism of GM1 ganglioside was initially hypothesized. The effects of the CV82 compound on the β-Gal enzyme were negligible, suggesting that the GM1 decrease is rather related to other yet unknown mechanisms that follow the direct action of CV82 on GCase. Additionally, CV82 did not decrease (instead, it increased) GM1 levels in fibroblasts carrying the R131C mutation, which can affect the active site of GCase, possibly leading to structural changes of the catalytic site that might prevent the interaction with CV82.

GCase defects are increasingly assimilated to a continuum of symptoms and signs since they can give rise to the three main GD phenotypes, to perinatal lethal GD and to at least two additional neurodegenerative diseases: Lewy body dementia and Parkinson’s disease [18,30]. The accumulation of GM1 ganglioside detected in the two different cell lines (bearing the L444P and RC131 homozygous mutations) could be useful to explain the complex scenario of the neurodegeneration observed in some severe forms of GD. It is well established that neurodegeneration in GM1 gangliosidosis is caused by GM1 storage [31,32]. The here-reported detection of GM1 storage materials also in GD cell lines from neuropathic patients suggests a possible role of this secondary storage material in GD pathogenesis.

Although gangliosides and glycosphingolipids are known to accumulate in many LSDs other than GD, the molecular mechanisms at the basis are mostly unknown [33,34]. The explanation of the metabolic pathways that can give rise to GM1-ganglioside accumulation in GD may provide a solid rationale for a better understanding of the physiopathology linked to GCase defects and of the corresponding phenotypes and be of relevance for other LSDs. If targeting GCase, since the enzyme involved in the primary storage accumulation has not yet led to any treatment for the neuropathic GD conditions, it could be worth trying to target the secondary accumulation of GM1 and find whether any beneficial effects are found in vivo. The results obtained here on the pharmacological treatment of patients’ cells carrying mutations that can affect either the stability of GCase (L444P) or the functionality of its active (R131C) might help in the future to establish personalized drug treatments for patients according to their mutations.

## 4. Materials and Methods

### 4.1. Cell Cultures

Fibroblast samples were obtained from the “Cell Line and DNA Biobank from Patients Affected by Genetic Diseases” (G. Gaslini Institute)—Telethon Genetic Biobank Network. The isolated skin fibroblasts from patients were cultured in Dulbecco’s Modified Eagle’s Medium with 10% foetal bovine serum and 1% penicillin/streptomycin antibiotics solution. The fibroblasts were characterized at the genetic and biochemical level. Their details are presented in Table 1. 

### 4.2. Inhibitory Activity towards Human Lysosomal β-Galactosidase (β-Gal)

The compound CV82 was screened at a 1 mM concentration towards β-Galactosidase (β-Gal) in leukocytes isolated from healthy donors (controls). Isolated leukocytes were disrupted by sonication, and a Micro BCA Protein Assay Kit (Sigma-Aldrich, St. Louis, MO, USA) was used to determine the total protein amount for the enzymatic assay, according to the manufacturer’s instructions. β-Gal activity was measured in a flat-bottomed 96-well plate. Compound-CV82 solution (3 μL), 4.29 μg/μL of leukocyte homogenate 1:10 (7 μL), and substrate 4-methylumbelliferyl β-d-Galactopyranoside (1.47 mM, 20 μL, Sigma-Aldrich) in acetate buffer (0.1 M, pH 4.3) containing NaCl (0.1 M) and sodium azide (0.02%) were incubated at 37 °C for 1 h. The reaction was stopped by the addition of sodium carbonate (200 μL; 0.5M, pH 10.7) containing Triton X-100 (0.0025%), and the fluorescence of 4-methylumbelliferone released by β-Gal activity was measured in a SpectraMax M2 microplate reader (λex = 365 nm, λem = 435 nm; Molecular Devices, San Jose, CA, USA). The percentage of β-Gal inhibition is given with respect to the control (Ctrl, without compound). Data are mean ± S.D. (*n* = 3). 

### 4.3. Pharmacological Chaperoning Activity

The fibroblasts wild type (WT) and with the p.Arg201His/p.Tyr83LeufsX8 and p.Arg201His/p.Ile51Asn mutations from juvenile GM1 patients were obtained from Meyer Children’s Hospital (Firenze, Italy). Fibroblast cells (20.0 × 10^4^) were seeded in T25 flasks with DMEM supplemented with foetal bovine serum (10%), penicillin/streptomycin (1%), and glutamine (1%) and incubated at 37 °C with 5% CO_2_ for 24 h. The medium was removed, and fresh medium containing the compound CV82 was added to the cells and incubated for 4 days. The medium was removed, and the cells were washed with PBS and detached with trypsin to obtain cell pellets, which were washed four times with PBS, frozen, and lysed by sonication in water. Enzyme activity was measured as reported above. Reported data are mean ± S.D. (*n* = 2). 

### 4.4. Confocal-Microscopy Analysis

Fibroblasts from the wild-type (WT) control and from GD patients carrying c.1226A>G (N370S)/RecNcil, c.1448T>C (L444P) and c.508C>T p. (R131C) mutations were plated in 12-well plates containing glass coverslips at a 30,000 cells/well density. Twenty-four hours after plating, the cells were fixed with 4% PFA, rinsed with PBS (+MgCl_2_ 0.5 mM, +CaCl_2_ 0.8 mM) and permeabilized with 0.075% Triton X. After rinsing with PBS and blocking with 4% BSA PBS, cells were incubated for 45 min with rabbit polyclonal anti-Glc 0.02% NaN_3_ (RAS0010, Glycobiotech, Kükels, Germany, diluted at 1:250) or 10 µg/mL Cholera Toxin B subunit—Fluorescein isothiocyanate conjugate (CTXb-FITC, Sigma-Aldrich Merck, Burlington, MA, USA) primary antibodies to stain GlcCer and GM1, respectively. After washing the coverslips with PBS, cells were incubated for 40 min with the secondary antibody Alexa 568 anti-rabbit (Invitrogen, diluted at 1:500). The coverslips were then washed with PBS and water and mounted on a glass slide. Cell imaging was performed on a Nikon Eclipse TE300 C2 confocal microscope (Nikon, Tokyo, Japan) equipped with a Nikon 100× immersion oil objective (Apo Plan, NA 1.4), with Melles Griot (Argon 488 nm) and Coherent (Sapphire 561 nm) lasers. Emission filters for imaging were 514/30 nm and 595/60 nm.

### 4.5. Flow-Cytometry Analysis

Fibroblasts from the wild-type (WT) control and from GD patients carrying c.1226A>G (N370S)/RecNcil, c.1448T>C (L444P) and c.508C>T p. (R131C) mutations were plated in 12-well plates at a 30,000 cells/well density. Twenty-four hours after plating, the cells were harvested and washed with PBS. A BD Cytofix/CytopermTM Fixation/Permeabilization Solution kit (BD Biosciences, Lake Franklin, NJ, USA) was used for fixation and permeabilization according to the manufacturer’s instructions. Then, fibroblasts were incubated with 10 µg/mL of rabbit polyclonal anti-Glc (diluted at 1:500) or 10 µg/mL of FITC-CTXb primary antibodies prepared in a 1× BD Perm/Wash buffer for 40 min at room temperature. After washing with 1× BD Perm/Wash buffer, cells were incubated with the secondary antibody Alexa 568 anti-rabbit (diluted at 1:500) for 30 min at room temperature. After washing with a 1× BD Perm/Wash buffer, the samples were analysed on a Accuri C6 flow cytometer (BD Biosciences, Lake Franklin, NJ, USA). Data were analysed using the free Flowing software (Version 2.5.1 downloadable at https://flowingsoftware.com/ or https://bioscience.fi/services/cell-imaging/flowing-software/ from Cell Imaging and Cytometry Core, Turku Bioscience Centre, Turku, Finland). The fibroblasts were identified by side-scattered (SSC) and forward-scattered (FSC) light. The GlcCer and GM1 were quantified by the median fluorescence intensity (MFI) of the cellular population labelled with the corresponding markers. For the treatment with CV82, fibroblasts from GD patients were plated in 12-well plates at a 20,000 cells/well density. Twenty-four hours after plating, the cells were washed with PBS and incubated with three concentrations of CV82 (10 µM, 1 µM, and 100 nM), previously solubilized in distilled H_2_O. After 4 days, cells were harvested, washed with PBS, fixed, and permeabilized as reported above and analysed with flow cytometry.

### 4.6. Statistical Analysis

The confocal-microscopy data were expressed as mean ± standard deviation (S.D.). Statistical significance was evaluated using Student’s test for both confocal-microscopy and flow-cytometry data, as we were interested in finding whether our method could discern unaffected and affected patients. A *p*-value lower than 0.05 was considered to be statistically significant. The single (*), double (**), and triple (***) asterisks refer to *p*-values lower than 0.05, 0.01, and 0.001, respectively. Statistical analysis was performed using the KaleidaGraph software (version 4.4.1). The minimum numbers of cells analysed are indicated in their respective legends.

## Figures and Tables

**Figure 1 molecules-29-00453-f001:**
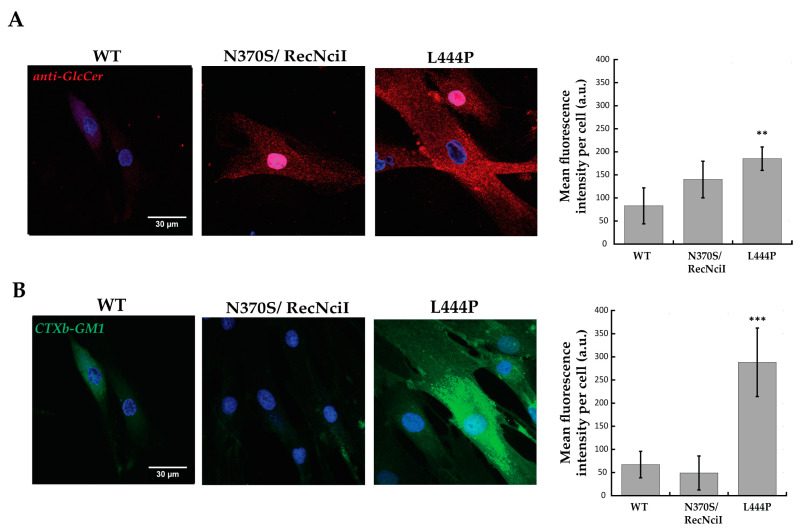
Thawed primary cultures of fibroblasts isolated from control wild type (WT) and GD patients carrying N370S/RecNcil and L444P mutations were cultured, fixed, permeabilized, labelled with anti-glucosylceramide and secondary Alexa 568 (**A**) or CTXb-FITC (**B**), and imaged with confocal microscopy. A significant increase in anti-Glc and CTXb-FITC fluorescence intensities was observed in fibroblasts carrying the L444P mutation with respect to the control, highlighting a rise in cellular content of GlcCer and GM1. Scale bar 30 μm. A total of >20 cells were analysed for each condition. Error bar S.D. Student’s *t*-test: ** *p* ≤ 0.01 and *** *p* ≤ 0.001.

**Figure 2 molecules-29-00453-f002:**
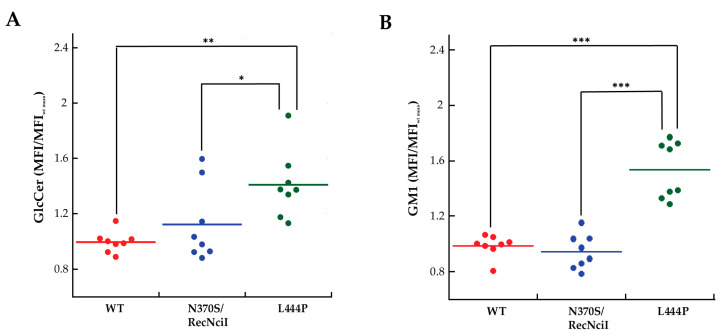
Flow-cytometry analysis of fibroblasts from WT and from GD patients carrying N370S and L444P mutations. Cells were fixed, permeabilized, and labelled with anti-GlcCer and secondary Alexa 647 antibodies (**A**) or CTXb-FITC (**B**). MFI/MFIWT mean values were obtained by dividing the MFI of a distribution by the mean MFI obtained from the controls. MFI/MFIWT mean values of anti-Glc (**A**) and CTXb-FITC (**B**) increased significantly in the fibroblasts carrying the L444P mutation with respect to the WT control and to fibroblasts carrying the N370S/RecNcil mutation, indicating a rise in content of GlcCer and GM1. A total of >4000 cells were analysed for each MFI dot. Student’s *t*-test: * *p* ≤ 0.05, ** *p* ≤ 0.01, and *** *p* ≤ 0.001. *n* ≥ 3 independent experiments.

**Figure 3 molecules-29-00453-f003:**
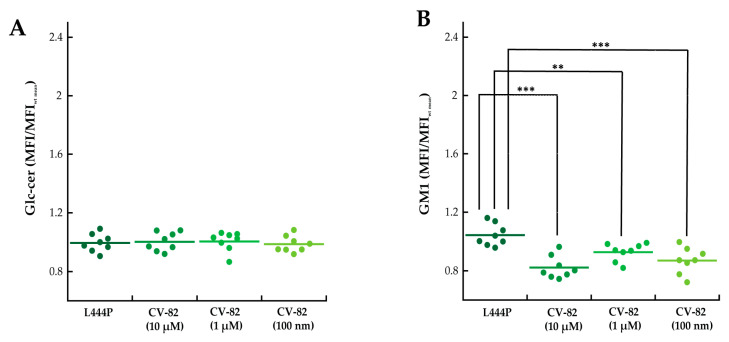
(**A**,**B**) Primary cultures of fibroblasts isolated from patients carrying the L444P mutation were incubated for 4 days with three different concentrations of CV82 (10 µM, 1 µM, and 100 nM), fixed, permeabilized, labelled with anti-glucosylceramide and secondary Alexa 647 antibodies (**A**) or CTXb-FITC (**B**), and analysed with flow cytometry. Treatment with 10 µM and 100 nM of CV82 led to a significant reduction in cellular content of GM1 but not GlcCer. Error bar S.D. Student’s *t*-test: ** *p* ≤ 0.01, and *** *p* ≤ 0.001. A total of >4000 cells were analysed for each MFI dot. *n* ≥ 3 independent experiments.

**Figure 4 molecules-29-00453-f004:**
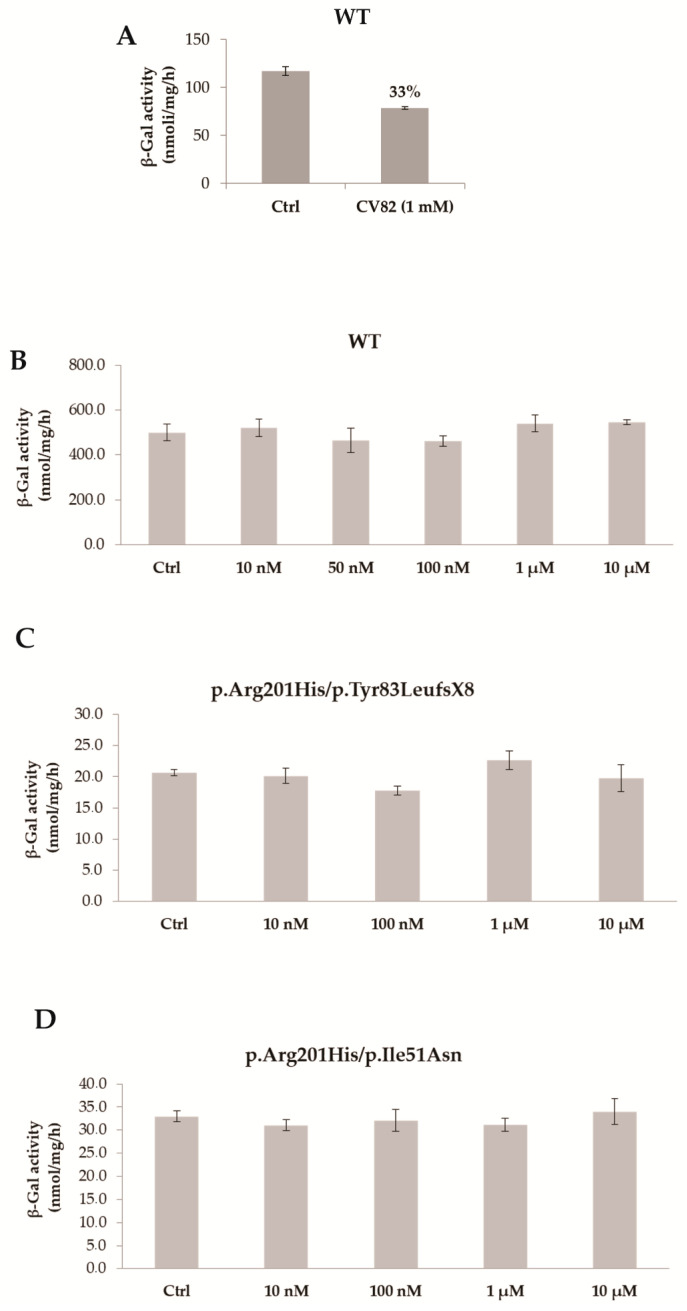
(**A**) Activity of β-Gal in the presence of compound CV82 in human leukocyte homogenates derived from healthy donors. The corresponding calculated percentage of inhibition is indicated above bar. (**B**–**D**) Fibroblasts WT (**B**) and bearing p.Arg201His/p.Tyr83LeufsX8 (**C**) and p.Arg201His/p.Ile51Asn (**D**) mutations were incubated without or with different concentrations of compound CV82. The β-Gal activity was determined in lysates from the treated fibroblasts. Reported data are mean ± S.D. (*n* = 2).

**Figure 5 molecules-29-00453-f005:**
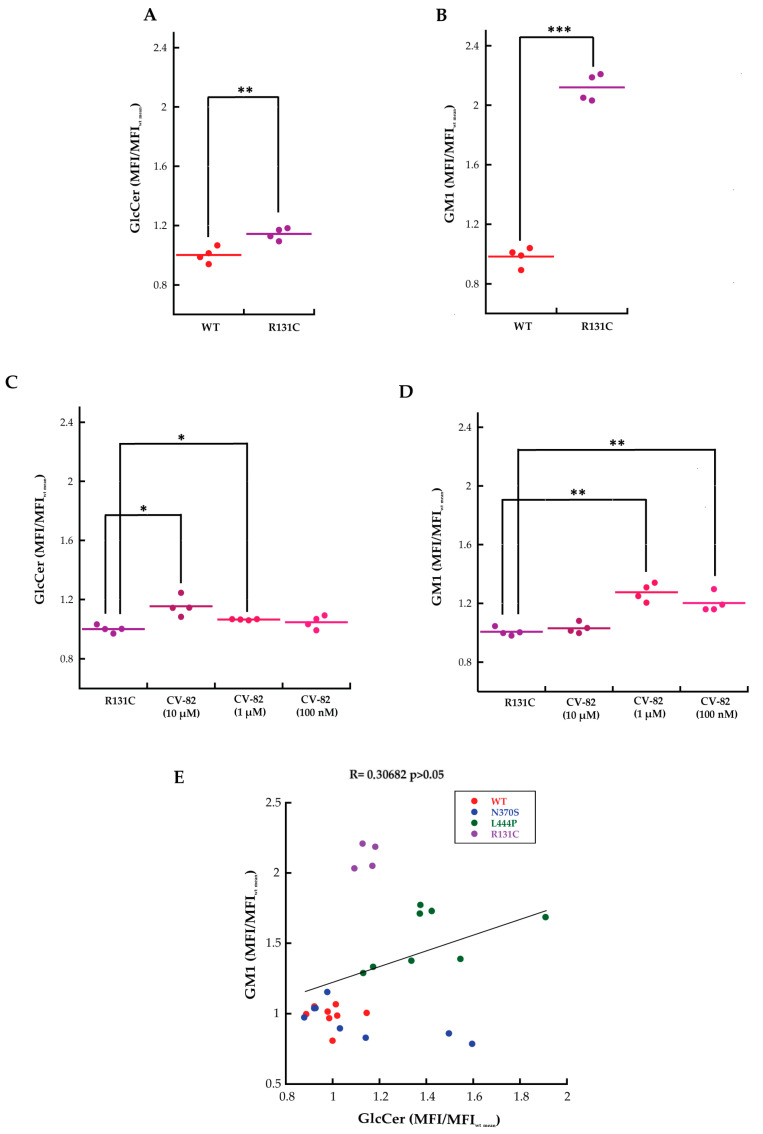
Flow-cytometry analysis of primary cultures of fibroblasts isolated from WT and GD patients carrying the R131C mutation. Fibroblasts were fixed, permeabilized, labelled with anti-glucosylceramide and secondary Alexa 647 (**A**) or CTXb-FITC (**B**). A total of >4000 cells were analysed for each MFI dot. MFI/MFIWT mean values of anti-Glc (**A**) and CTXb-FITC (**B**) significantly increased in fibroblasts of patients carrying R131C mutation compared with WT, indicating an increase in GlcCer and GM1 storages. Treatment of primary cultures of fibroblasts isolated from patients for 4 days with different concentrations of CV82 led to a significant increase in cellular content of both GlcCer (**C**) and GM1 (**D**). Error bar S.D. Student’s *t*-test: * *p* ≤ 0.05, ** *p* ≤ 0.01, and *** *p* ≤ 0.001. (**E**) Relation between GlcCer and GM1 levels in WT and patients carrying N370S/RecNcil, for L444P and R131C mutations. A total of >4000 cells were analysed for each MFI dot. *n* ≥ 3 independent experiments.

**Table 1 molecules-29-00453-t001:** Molecular, genetic, and enzymatic characterization of GD-derived fibroblast cell lines.

GD-Derived Fibroblasts	Mutated Gene	Nucleotide Changes	Amino Acid Changes	β-Glucocerebrosidase Activity in Fibroblast (n.v.: 120–400 nmol/mg/h)
N370S	*GBA* NM_000157.1	c.1226A>G/ RecNcil	p.N370S/ RecNcil	44.4
L444P	*GBA* NM_000157.1	c.1448T>C/ c.1448T>C	p.L444P/ p.L444P	11.8
R131C	*GBA* NM_000157.1	c.508C>T/ c.508C>T	p.R131C/ p.R131C	1.7

**Table 2 molecules-29-00453-t002:** Chemical structure of the trivalent compound CV82 and its biological property vs. GCase.

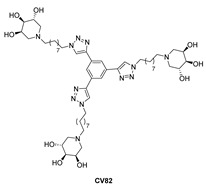	**GCase Inhibition (%) ^(a)^**	**IC_50_** **(µM) ^(a)^**	**Mutated GCase Activity Rescue ^(a)^**	
			**N370S/RecNcil**	**L444P/L444P**
	100	7 ± 1 competitive inhibition (K_i_ = 3.1 ± 0.2 μM)	1.21 at 10 μM	1.07 at 10 μM

^(a)^ Data taken from Ref. [17].

## Data Availability

The data presented in this study are available in article.

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
