# Peer review of "Identification of GM1-Ganglioside Secondary Accumulation in Fibroblasts from Neuropathic Gaucher Patients and Effect of a Trivalent Trihydroxypiperidine Iminosugar Compound on Its Storage Reduction"

_molecules, 2024, doi:10.3390/molecules29020453_

Round 1

Reviewer 1 Report

Comments and Suggestions for Authors

This manuscript Identified the GM1 ganglioside secondary accumulation in fibroblasts from neuropathic Gaucher patients, and investigated the effects of trivalent trihydroxypiperidine iminosugar compound CV82 on its storage reduction. Some modifications should be made to improve the quality of this manuscript.

1. Introduction: what's the practical or theorical significance of this study? Add related details.

2.  Results: strongly suggest to re-wtite this section because of the following points.

2.1. Each sub-title of this section looks like a sentence rather than a formal title. Revise it.

2.2. Too many active voices were used, such as "we tested/monitored/evaluated ...", which should be modified to positive voices in grammer (... was/were tested/monitored/evaluated).

2.3. "2.5. Figures and Tables" can naver be a independent part of this section. The figures and tables shall be devided into correct places. Revise them.

2.4. Table 2: How did you detect the IC50 value? By which method? I didn't see any detailed information about this. Add the description in the Materials and Methods section.

2.5. Also Table 2: strongly suggest to exhibit the dose-response curve of CV82 inhibiting GCase with regression model and R2.

3. Where is the Conclusion section? It is not necessary for the articles published on this journal?

4. Materials and Methods: What is the full name of CLSM? And FITC? LSCM? Add associate statements.

5. L399: "Statistics" is usually "Statistical analysis".

6. References: some cited literatures are too old, even published within the past century (before 2000 A.D.).

Comments on the Quality of English Language

Moderate language editing is required, e.g., too many active voices should be changed to positive voices in writing English.

Author Response

Reply to Reviewer 1

  1. Introduction: what's the practical or theorical significance of this study? Add related details.

Our results on the secondary accumulation of GM1 are important for the characterization of the downstream pathways in GD and potentially pave the way for testing β-galactosidase as alternative therapeutic target for neuronopathic variants of GD with unmet clinical need. We added this sentence to the Introduciton as requested by the reviewer.

  1. Results: strongly suggest to re-wtite this section because of the following points.

2.1. Each sub-title of this section looks like a sentence rather than a formal title. Revise it.

We re-phrased each sub-title of the results section as requested by the reviewer.

2.2. Too many active voices were used, such as "we tested/monitored/evaluated ...", which should be modified to positive voices in grammer (... was/were tested/monitored/evaluated).

We decreased by approximately the half the use of ‘we…etc.’ and modified the text according to the referee’s suggestion.

2.3. "2.5. Figures and Tables" can naver be a independent part of this section. The figures and tables shall be devided into correct places. Revise them.

Section 2.5 was deleted as requested by the reviewer and figures, legends and tables were added after their corresponding citation in the text.

2.4. Table 2: How did you detect the IC50 value? By which method? I didn't see any detailed information about this. Add the description in the Materials and Methods section. & 2.5. Also Table 2: strongly suggest to exhibit the dose-response curve of CV82 inhibiting GCase with regression model and R2.

Thank you for noting, indeed it was misleading, since the biological evaluation of CV82 vs GCase as inhibitor and pharmacological chaperone has been previously reported (ChemBioChem 2022, e2022000779; the acronym CV82 corresponds at compound 12).

We therefore indicated the Reference as Table footnote [a].

  1. Where is the Conclusion section? It is not necessary for the articles published on this journal?

The guidelines for the authors of this journal state that the Conclusion section is not mandatory. The conclusions are reported in the last paragraph of the Discussion section.

  1. Materials and Methods: What is the full name of CLSM? And FITC? LSCM? Add associate statements.

We replaced CLSM (confocal laser scanning microscopy) simply with ‘confocal microscopy’. We reported the full name of FITC (Fluorescein isothiocyanate) in M&M section. We replaced LSCM (laser scanning confocal microscope) simply with ‘confocal microscope’.

  1. L399: "Statistics" is usually "Statistical analysis".

We changed the sub-paragraph to ‘Statistical analysis’ as suggested by the referee.

  1. References: some cited literatures are too old, even published within the past century (before 2000 A.D.).

We have checked the references and we think that, although old, the listed references are needed. In the case of rare diseases the references regarding some mutations as well as data on accumulation of secondary storage materials are very few, and often old. We do not agree that the time of publication should be a reason to skip a citation.

Comments on the Quality of English Language

-Moderate language editing is required, e.g., too many active voices should be changed to positive voices in writing English.

See response to point 2.2

Reviewer 2 Report

Comments and Suggestions for Authors

I was very pleased to review the manuscript "Identification of GM1 ganglioside secondary accumulation in fibroblasts from neuropathic Gaucher patients and effect of a trivalent trihydroxypiperidine iminosugar compound on its storage reduction". This paper nicely addresses the problem of secondary ganglioside storage, which contributes to the pathogenesis of neuronopathic Gaucher disease, and demonstrates a promising therapeutic approach. The study on fibroblasts was well designed and conducted, and the results are presented clearly. I have only one suggestion, please add about the use of piperidine compounds as medicinal products and data about the safety in humans. 

Author Response

Reply to Reviewer 2

-I have only one suggestion, please add about the use of piperidine compounds as medicinal products and data about the safety in humans. 

We thank the reviewer for addressing the point on the use of trihydroxypiperidine iminosugars as potential therapeutic products. For this reason, we added a reference relating to the interesting biological activity of piperidine iminosugars as recently reviewed by Simone and coworkers. Moreover we emphasized that the first commercially approved PC for Fabry is a piperidine iminosugar compound and reported a reference about the pharmacokinetic, pharmacodynamic, and tolerance data of this drug.

To better address this issue, we modified the Text as follows:

Trihydroxypiperidine iminosugars, carbohydrate analogues in which a nitrogen atom replaces the endocyclic oxygen, have shown interesting properties as anti-viral, anti-bacterial, anti-diabetes or immunosuppressant agents [Prichard, K.; Campkin, D.; O’Brien, N.; Kato, A.; Fleet, G.W.J.; Simone, M.I. Biological activities of 3,4,5-trihydroxypiperidines and their N- and O- derivatives. Chem. Boil. Drug Des. 2018, 92, 1171–1197], and more recently also as PCs for LSDs [10,11]. In the past twenty years, other piperidine iminosugars have been investigated to this aim, such as 1-deoxygalactonojirimycin (DGJ), which was the first oral PC commercially approved for Fabry Disease (another LSD) in Europe (Migalastat, Galafold, Amicus Therapeutics) [McCafferty, E.H., Scott, L.J. Migalastat: A Review in Fabry Disease. Drugs 2019, 79, 543–554]. However, no PC for Gaucher disease has yet reached the market.

Round 2

Reviewer 1 Report

Comments and Suggestions for Authors

As the authors have modified the manuscript carefully point by point based on the reviewers' comments and suggestions, I suggest that this manuscrip can be accepted in the current form.